# Biochemical Profile and In Vitro Therapeutic Properties of Two Euhalophytes, *Halocnemum strobilaceum* Pall. and *Suaeda fruticosa* (L.) Forske., Grown in the Sabkha Ecosystem in the Algerian Sahara

**DOI:** 10.3390/molecules28083580

**Published:** 2023-04-19

**Authors:** Noura Gheraissa, Ahmed Elkhalifa Chemsa, Nezar Cherrada, Ebru Erol, Eman Ramadan Elsharkawy, Djilani Ghemam-Amara, Soumeia Zeghoud, Abdelkrim Rebiai, Mohammed Messaoudi, Barbara Sawicka, Maria Atanassova, Maged S. Abdel-Kader

**Affiliations:** 1Laboratory of Biodiversity and Application of Biotechnology in Agriculture, El Oued University, El Oued 39000, Algeria; nouragherr@gmail.com (N.G.); khalifa-chemsa@univ-eloued.dz (A.E.C.); cherrada-nezar@univ-eloued.dz (N.C.); 2Department of Cellular and Molecular Biology, Faculty of Natural Science and Life, El Oued University, El Oued 39000, Algeria; 3Department of Biology, Faculty of Natural Science and Life, El Oued University, El Oued 39000, Algeria; djilani-ghemamamara@univ-eloued.dz; 4Department of Analytical Chemistry, Faculty of Pharmacy, Bezmialem Vakif University, İstanbul 34093, Türkiye; 5Department of Chemistry, Faculty of Science, Northern Border University, Arar 73213, Saudi Arabia; 6Laboratory of Biology, Environment and Health, El Oued University, El Oued 39000, Algeria; 7Chemistry Department, Faculty of Exact Sciences, University of El Oued, El Oued 39000, Algeriamessaoudi2006@yahoo.fr (M.M.); 8Nuclear Research Centre of Birine, Ain Oussera 17200, Algeria; 9Department of Plant Production Technology and Commodities Science, University of Life Science in Lublin, Akademicka 15 Str., 20-950 Lublin, Poland; 10Nutritional Scientific Consulting, Chemical Engineering, University of Chemical Technology and Metalurgy, 1734 Sofia, Bulgaria; 11Department of Pharmacognosy, College of Pharmacy, Prince Sattam Bin Abdulaziz University, Al-Kharj 11942, Saudi Arabia; 12Department of Pharmacognosy, Faculty of Pharmacy, Alexandria University, Alexandria 21215, Egypt

**Keywords:** *Halocnemum strobilaceum*, *Suaeda fruticosa*, euhalophytes, chott salt, biochemical profile

## Abstract

This study reports the biochemical profile and in vitro biological activities of the aerial part of two shrubs: *Halocnemum strobilaceum* and *Suaeda fruticosa*, a halophytes species native to saline habitats. The biomass was evaluated by determining its physiological properties and approximate composition. Hydro-methanolic extracts from *Halocnemum strobilaceum* and *Suaeda fruticosa* have been investigated for the inhibition of bacterial growth, the protection of proteins (albumin) from denaturation, and cytotoxicity to hepatocellular carcinomas (Huh-7 and HepG2). Their antioxidant activity was evaluated by five tests, including one that examined their ability to inhibit hydrogen peroxide (H_2_O_2_)-induced hemolysis. The profile of their phenolic compounds was also determined. These two euhalophytes had a high moisture content, high levels of photosynthetic pigments, elevated levels of ash and protein, low oxidative damage indices, MDA (Malondialdehyde) and proline, and low lipids levels. Their content was also characterized by a moderate acidity with good electrical conductivity. They contained abundant levels of phytochemicals and varied phenolic contents. Reverse phase high performance liquid chromatography (RP-HPLC) analysis revealed the presence of caffeic acid, *p*-coumaric acid, rutin, and quercetin in both plant extracts. On the pharmaceutical level, the two euhalophytes had anti-inflammatory, antibacterial, antioxidant, and cytotoxic properties, and therefore it was recommended to isolate and identify biologically active compounds from these plants and evaluate them in vivo.

## 1. Introduction

Chott salt or Sabkha are salt lakes, and they are one of the inland aquatic ecosystems that are related to the marine environment. Salt lakes can be permanent or temporary water bodies with a salt content that exceeds 3 g/mL. They have a dry climate with a high rate of evaporation and scarce precipitation and a soil that is rich in minerals. Salt lakes have a wide geographical extension that includes all continents [1,2]. Chotts are considered lowlands belonging to the wetlands and are known to have a unique ecological role as they contribute to biodiversity events. Chotts can also have an economic role; for example, they constitute the most important pasture for camels and therefore generate economic values from tourist activities. Two of the largest Chotts in North Africa are Chott Melghir in Algeria and Chott Merouane in Tunisia [1]. These environments have their own biodiversity, represented by some algae and certain animals (such as crustaceans, subclass copepods, some terrestrial invertebrates, and reptiles) which are capable of withstanding these harsh conditions and able to flourish and grow within them [3]. Chotts can also form a regional food web for some migratory birds, and vegetation cover is limited to halophytic species [4].

A halophyte is a salt-tolerant flowering plant that can adapt and grow in saline habitats [5]. They are distinct from other plants because they use more than one mechanism to regulate salt and avoid salinity. These mechanisms depend on their anatomical, morphological and physiological characteristics to tolerate and/or resist such stress [6]. These mechanisms are used to maintain basal growth under severe salinity conditions, by protecting the photosynthetic system by chlorophyll synthesis and promoting or inhibiting carotenoids, through regulating osmotic modulation to reduce water deficit, and by lowering oxidative stress by producing antioxidant products, such as phenolic compounds and flavonoids [7]. According to the physiology of salt tolerance, halophytes are classified into three classes: euhalophytes, recretohalophytes, and salt exclusion halophytes. Euhalophytes can dilute salt within their succulent leaves and stems as a form of high salinity tolerance. Recretohalophytes have salt glands which allow the direct excretion of salt from the plant. Salt exclusion halophytes exclude salt by eliminating most of the sodium (Na^+^) and chlorine (Cl^−^) in the soil [6]. Halophytes are considered vegetables due to their organoleptic properties and salty taste, and medicinal plants due to the presence of bioactive compounds. However, the nutritional/chemical composition of halophytes has not been studied in depth [8].

Conducting nutritional analysis in a study concerned with determining the therapeutic properties of medicinal plants provides information regarding the safe usage of these plants by the consumer (human and animal), and also confirms their therapeutic and nutritional role. In this study, the two most important halophytes found in saline desert environments, *H. strobilaceum* and *S. fruticosa* in El Oued, Algeria, were studied, and they belong to the same family: *Chenopodiaceae*. Studying these halophytes can help to understand the effect that harsh conditions have on the chemical content of plants, by measuring the content of secondary and primary metabolites. By defining their therapeutic role, we seek to obtain biologically active compounds that can be exploited in the pharmacological and/or nutritional field. This study is the first of its kind to determine the physiological and physicochemical properties of the aerial part of *H. strobilaceum* and *S. fruticosa*, in particular. In addition, the study determines the uniqueness of the phytochemical interest of these two halophytes, *H. strobilaceum* and *S. fruticosa*, specifically growing in the Algerian desert, by determining the largest possible amount of their phytochemical content and also by identifying a spectrum of their biological activities. These include antioxidant activity determined by six assays, antibacterial activity of six pathogenics, in vitro anti-inflammatory activity, and the cytotoxicity activity of two types of liver cancer cells (Huh-7 and HepG2).

## 2. Results

### 2.1. Physiological Characteristics

There was no significant difference (*p* > 0.05) in the content of proline and MDA between *H. strobilaceum* and *S. fruticosa* (Table 1), but there was a significant difference (*p* < 0.05) in moisture percentage and pigment content (chlorophyll a, chlorophyll b, and carotene).

### 2.2. Physicochemical Parameters

The SEM-EDX analysis showed that the two plant extracts contained the same type of oxides (CaO, Na_2_O, SiO_2_, Al_2_O_3_, and MgO) and elements (Table 2).

*H. strobilaceum* had the highest ash content, pH, electrical conductivity, and the lowest carbohydrate content (Table 3). No significant difference in lipid and protein content was observed between *H. strobilaceum* and *S. fruticosa* (*p* > 0.05).

### 2.3. Phytochemical Analysis

The total phenolic contents of the two hydro-methanolic extracts are shown in Table 4. The difference in the content of phenolic compounds, with the exception of anthocyanins, between *H. strobilaceum* and *S. fruticosa*, were statistically significant (*p* < 0.05).

Table 5 displays the quantity of the phenolic compounds identified in the plant extracts. The main compound identified in the *H. strobilaceum* extract was quercetin (207.16 µg/100 mg), followed by chlorogenic acid (85.77 µg/100 mg) and rutin (84.42 µg/100 mg). In the *S. fruticosa* extract, rutin (369.56 µg/100 mg), quercetin (93.69 µg/100 mg), and vanillic acid (20.30 µg/100 mg) were identified as the main compounds.

Figure 1 shows the peaks of the phenolic compounds identified by RP-HPLC. Six compounds, out of a total of 63, were identified in the *H. strobilaceum* extract. The *S. fruticosa* extract contained 62 compounds, of which seven out of the nine reference compounds were identified.

### 2.4. Biological Activity

#### 2.4.1. Assessment of Antioxidant Efficiency

The IC_50_, Hly_50_ or EC_50_ values for the various assays are listed in Table 6. In all values, significant differences were found between samples (*p* < 0.05). In most assays, the superiority of the control was positive, followed by *H. strobilaceum* extract and then the *S. fruticosa* extract.

#### 2.4.2. Assessment of Antibacterial Activity

According to the results shown in Table 7 and the statistical analysis (*p* < 0.05), it is evident that the two studied extracts and the positive control (Gentamicin) showed activity against the six bacteria strains tested. Three Gram-positive strains had poor sensitivity to gentamicin. While *S. typhimurium* and *P. aeruginosa* had strong sensitivity towards Gentamicin, it had no effect against *E. coli*, while the six strains had weak or moderate sensitivity to the two extracts, *H. strobilaceum* and *S. fruticosa*. Where *H. strobilaceum* extract recorded the best activity against *E. coli* (15.3 ± 3.2), it also had the lowest activity against *P. aeruginosa* (8 ± 1.7).

#### 2.4.3. Evaluation of Anti-Inflammatory Efficacy

The anti-inflammatory activity results shown in Figure 2 showed that the extracts of *H. strobilaceum* and *S. fruticosa* significantly decreased the rate of inhibition of protein denaturation. The statistical study showed that there were no significant differences (*p* > 0.05) between the two extracts and the positive certificate (Diclofenac sodium).

#### 2.4.4. Assessment of Anticancer Potential

The percentage of viability of the Huh-7 and HepG2 cells is shown in Figure 3. It should be noted that the extracts have moderate cytotoxicity activity because the percentage of dead cells in the *H. strobilacum* extract was more than 20% even though the extract concentrations used to treat the cells did not exceed 100 µg/mL. The percentage of dead cells observed after treatment with the *S. fruticosa* extract was more than 10%. This efficacy was equivalent to the efficacy of the positive control (Epiphany), according to the statistical analysis, which proves that there is no significant difference between the studied samples (*p* > 0.05).

## 3. Discussion

The high moisture content observed in the *H. strobilaceum* and *S. fruticosa* extracts indicated that they are salt-tolerant plants that are able regulate osmotic potential [9] and store water inside their tissues. Thus, they maintain their physiological growth in such harsh conditions [10]. In addition to being euhalophytes, Glenn and O’Leary [11] stated that this type of halophyte has a tight coordination when producing a constant osmotic potential gradient within their cellular tissues relative to the soil solution (high salinity), which ensures its higher water content compared to other halophytes. Qasim et al. [12] also found, through their estimation of the nutritional value of five euhalophytes including *S. fruticosa*, that their moisture content ranged between 68.58 and 80.21%. The difference in the level of stored water in *H. strobilaceum* and *S. fruticosa* can be attributed to the fact that *H. strobilaceum* has succulent leaves while *S. fruticosa* has succulent aerial parts. Since the primary role of the succulent tissue is to store water reserves, the water content in the succulent organs may reach 90–95% [5].

These two plants also contain a high content of pigments, especially chlorophyll a. The ability of the plant to tolerate biological stresses such as salinity and drought depends on an increase in the thickness of the mesophyll layer, which leads to an increase in the content of chlorophyll and ultimately in the photosynthetic capacity of the plant [13,14]. The low levels of proline and MDA in the plants indicate a low rate of oxidative stress [15].

The nutrient content of the plant reflects the content of the soil in which it is grown. The two halophytes, *H. strobilaceum* and *S. fruticosa*, are a promising source of some essential trace minerals, along with some toxic heavy metals accumulated from their environment (salt marshes). When analyzed in the scanning electron microscope, these two plants were found to be devoid of most nutrients, except for calcium (Ca), which was present in adequate proportions, and magnesium (Mg), sulfur (S), and aluminum (Al) which were present in weak proportions. This is because saline-sodic soils lack nutrients such as iron (Fe), zinc (Zn), copper (Cu), and manganese (Mn) but are rich in inorganic ions such as chlorine (Cl^−^), sodium (Na^+^), potassium (K^+^), and calcium (Ca^+2^) [16,17]. For silicon (Si), its presence was evidence of mineral stress such as phosphorus deficiency, or mineral toxicity with Mn and/or Al [18]. This is in line with the findings of this investigation, where phosphate was completely absent in the two studied plants.

As for the oxide content of the two plants, desert plants are known to produce different quantities and types of oxides. For example, the formation of calcium oxides, which are present in cells in the form of crystalline deposits, is a mechanism for regulating calcium content in tissues [19].

The ash content reflects mineral and inorganic compounds such as mineral salts, which are essential for the plant. The ash content of traditional grass is about 5 to 10% [20], but halophytes in general have a higher ash content than other plants [21]. This is consistent with the findings of this study, where the ash content ranged from 20 to 30%. It is also consistent with the study by Maatallah Zaier et al. [8] involving three halophytic plants (*Arthrocnemum indicum*, *Halocnemum strobilaceum*, and *Suaeda fruticosa*) grown in Sabkha Sidi El Hani in Tunisia.

The reason for this might be that halophytic plants adapt to salinity by osmotic balance by increasing the accumulation of minerals [22]. On the other hand, some halophytes adopt the strategy of accumulating salts obtained from the soil solution inside the tonoplast, as a response to high salinity climatic conditions. This is to ensure their water requirements [23]. The difference in ash content observed between *H. strobilaceum* and *S. fruticosa* may be because the mineral content in plants depends on several variables, including plant type, soil amendments, weather, and soil type [24]. The pH was roughly moderate, between 7.5 ± 0.005 and 6.15 ± 0.09, although the two plants contained accumulated salts. This was because *Chenopodiaceae*, most of the species with a C4 metabolic pathway, are characterized by the accumulation of the organic acids malate and/or aspartate in the vacuoles, and this has the greatest effect on acidification [25,26]. The increase in conductivity values is evidence that the plant has an accumulation of solutes (Osmolytes) in the context of providing its water needs, as mentioned earlier.

With regard to macromolecule content, proteins make up the largest percentage in the *H. strobilaceum* and *S. fruticosa*, followed by carbohydrates and then lipids. This finding is consistent with the study by Qasim et al. [12], where the proportion of protein was higher than the proportion of carbohydrates in the five euhalophytes studied. However, in their study of ten medicinal plants, Radha et al. [27] observed that carbohydrates were the most abundant macromolecule in the extracts. The difference between the studies is thought to be due to the nature of the plant species or the estimating method used. The high content of proteins was also associated with the absence of nitrogen in SEM-EDX analyses. It is possible to explain this in view of the fact that the halophytes adopt a strategy to reduce growth rate through several mechanisms, including increased protein synthesis, which leads to increased nitrogen fixation [28].

Through physicochemical and physiological characteristics, it is clear that the two desert plants, *H. strobilaceum* and *S. fruticosa*, have important nutritional content, which may suggest their exploitation as a food or pastoral source. They can also be food products, additives with sensory properties to foods, and functional foods that have a desirable physiological effect beyond basic nutrition, dietary supplements, or nutraceuticals.

Principal results showed that the hydro-methanolic extract of *H. strobilaceum* contains 24.97 ± 0.09 µg equivalent of gallic acid as its polyphenolic content, and 12.17 ± 0.16 and 5.43 ± 0.06 µg equivalents of quercetin as the total flavonoid and flavanols content, respectively (Table 4). It showed more than 1.5 μg equivalent of cyanidin-3-glucoside as the anthocyanin content, 6.23 ± 0.24 µg of gallic acid as hydrolysable tannins, and 3.99 ± 0.09 µg equivalent of catechins as condensed tannins. This is consistent with the results from Handoussa et al. [29], who showed that the ethyl acetate extract of the aerial part of *H. strobilaceum* had a value of 29.42 mg GAE/g DW for phenolic contents.

The *S. fruticosa* extract contained phenols (47.38 ± 0.16 µg GAE/mg), flavonoids (14.57 ± 0.12 µg QE/mg), flavanols (6.70 ± 0.16 µg QE/mg), anthocyanins (1.17 ± 0.47 mg C-3-GE/mg), and hydrolysable (8.81 ± 0.32 µg GAE/mg) and condensed tannins (4.68 ± 0.25 µg CE/mg). These results are consistent with the findings from the Chekroun-Bechlaghem et al. [30] study, which collected samples from the same species from Sabkha in Northwest Algeria (47.73 ± 1.17 GAE/mg of phenolics, 4.27 ± 0.12 µg CE/mg of flavonoids, 7.76 ± 0.28 CE/mg of condensed tannins, and 1.75 ± 0.13 QE/mg of flavanols), and from a study by Qasim et al. [12] where they found that the extract of *S. fruticosa* grown in the coastal areas of Pakistan contained 46.54 ± 4.32 µg GAE/mg of phenolics, 21.43 ± 4.32 µg QE/mg of flavonoids, 8.71 ± 0.76 µg TAE/mg of tannins, and 15.76 ± 1.43 µg CE/mg of proanthocyanidins. Similarities in findings were also observed between this study and the study performed by Oueslati et al. [31], who analyzed the same species grown in Sabkha Kairouan in Tunisia and found 31.76 µg GAE/mg of polyphenols, 26.2 µg CE/mg of flavonoids, and 1.5 µg CE/mg of tannins in the extracts.

Through the results of quantification (Table 4) and quantification results of previous studies on *H. strobilaceum* and *S. fruticosa* mentioned earlier, we note that there is a convergence in the results. This may be due to the similarity of the environments in which these plants grow, which are saline habitats.

The quantitative and qualitative content of phenolic compounds is due to their genetic nature, as well as the environmental conditions. For example, in this study, hydrolysable and condensed tannins represented about 30 to 40% of the total phenolic compounds in the extracts; this is considered a remarkably high value. This can be attributed to the fact that wild plants produce tannins as a strategy to deter predators that attack them; tannins have an astringent flavor that animals do not like [32].

RP-HPLC analysis indicated some quantitative and qualitative phytochemical similarities between *H. strobilaceum* and *S. fruticosa*, which were quercetin, rutin, *p*-coumaric acid, caffeic acid, and vanillic acid, as well as the absence of Naringin. AbdelRazek and his team [33] mentioned that the aerial part of the *H. strobilaceum* plant contains many phenolic compounds, including quercetin, coumarin, and caffeic acid. Qasim et al. [12], through quantitative RP-HPLC analysis, determined the identity of each of the following phenolic compounds, quercetin, kaempferol, caffeic acid, chlorogenic acid, catechin, and gallic acid, in the aqueous-methanolic extract of *S. fruticosa*.

Based on the comparison between Table 5 and previous studies (see Appendix A), it can be observed that both *H. strobilaceum and S. fruticosa* contain significant amounts of phenolic acids and flavonoids, which is consistent with the findings of previous studies on other plants of the *Chenopodiaceae* family. For example, Gheraissa et al. [34] found that *Bassia muricata* contains high levels of chlorogenic acid, gallic acid, *p*-coumaric acid, vanillic acid, and vanillin, which are also present in *H. strobilaceum* and *S. fruticosa*. Additionally, El-Beltagi et al. [35] reported the presence of gallic acid, *p*-coumaric acid, and quercetin in *Beta vulgaris* L. root, which were also detected in *H. strobilaceum* and *S. fruticosa*.

In terms of the quantities of individual phenolic acids and flavonoids, the content of chlorogenic acid in *H. strobilaceum* (85.77 µg/100 mg ED) was lower than that reported in *Bassia muricata* (288 µg/100 mg ED) by Gheraissa et al. [34], while the content of *p*-Comuaric acid in *S. fruticosa* (1.63 µg/100 mg ED) was higher than that reported in *Beta vulgaris* L. root (0.74 µg/100 mg ED) by El-Beltagi et al. [35]. On the other hand, the amount of rutin in *S. fruticosa* (367.56 µg/100 mg ED) was much higher than that reported in other plants of the *Chenopodiaceae* family, such as *Bassia muricata* (32 µg/100 mg ED) by Gheraissa et al. [34] and *Beta vulgaris* L. root (not detected) by El-Beltagi et al. [35].

Upon comparing the compounds present in the *Chenopodiaceae* family but not detected in the current study of *H. strobilaceum* and *S. fruticosa*, it can be observed that *Bassia muricata* contains gallic acid, vanillin, and naringin, which are absent in *H. strobilaceum*, and chlorogenic acid and naringin, which are absent in *S. fruticosa* extracts. These compounds have been reported to possess antioxidant and anti-inflammatory properties [34], indicating that the absence of these compounds in the studied plants may limit their potential as a source of these bioactive compounds.

*Beta vulgaris* L. (root) contains catechol, ferulic acid, o-Coumaric acid, cinnamic acid, myricetin, neringenin, coumarin acid, resorcinol, and naphthaline, which are also absent in both *H. strobilaceum* and *S. fruticosa* extracts. These compounds have been reported to possess various biological activities, such as antioxidant, antibacterial, and anticancer activities [35,36,37], suggesting that the studied plants may not have the same potential as *Beta vulgaris* L. (root) in terms of their bioactive compound content.

The absence of these compounds in *H. strobilaceum* and *S. fruticosa* extracts suggests that they may not be a rich source of these particular bioactive compounds. The possible explanation for the absence of certain compounds in the current study compared to previous studies is that different methods of analysis can have different sensitivities and selectivities for certain compounds. Additionally, the sample preparation and extraction techniques can also affect the extraction efficiency of certain compounds, which can further contribute to the differences in results between studies.

It is also worth noting that differences in environmental conditions, such as soil type, water availability, and temperature, can affect the biosynthesis of secondary metabolites in plants. Therefore, the absence of certain compounds in the current study could also be attributed to differences in growing conditions or genetic variability between plant samples. However, it is important to note that the studied plants may contain other beneficial compounds that were not analyzed in the present study, and further investigation is needed to fully explore their potential as a source of bioactive compounds. Overall, the results of the current study suggest that *H. strobilaceum* and *S. fruticosa* grown in the Sabkha ecosystem of the Algerian Sahara have high levels of phenolic compounds, which may have potential health benefits. The findings of previous studies on the phenolic content of other plants in the *Chenopodiaceae* family, as reported in previous studies (see Appendix A), provide valuable context for interpreting the results of the current study and suggest that the observed phenolic profile in *H. strobilaceum* and *S. fruticosa* is consistent with that found in other plants of the family. These findings could be significant for future research that aims to explore the potential use of these plants for medicinal or industrial purposes.

In the DPPH^•^ assay, the hydro-methanolic extract of the aerial part of *H. strobilaceum* provided an IC_50_ value of 81.7 ± 0.64 µg/mL. This result converges with the result of the Saada et al. [38] study that observed IC_50_ values of 61 and 107.5 µg/mL for the apolar and polar fractions of the aerial parts of *H. strobilaceum* from south Tunisia, respectively. It is possible that the convergence of the results from these two studies is due to the similarity in geographical location and environmental conditions, such as soil and climate, whereas the IC_50_ value of the *S. fruticosa* extract observed in this study (118.8 ± 1.46 µg/mL) supported the finding from a study performed by Naija et al. [39], which reported the hydro-methanolic extract of the same plant to be 100 µg/mL. However, the IC_50_ value obtained for the hydro-acetonic extract (37 µg/mL) in a study by Oueslati et al. [40] was lower than the two above-mentioned studies. The discrepancy between the studies may be due to different extraction methods and the type of solvent used; these variables significantly influence the qualitative and quantitative properties of antioxidants [41].

For the OH^•^ assay, the IC_50_ values exceeded 1200 µg/mL, while for ascorbic acid, the IC_50_ value did not exceed 90 µg/mL. The difference in the ability to scavenge free radicals, in general, can be attributed to the structural features of the active components of the samples, which determine their ability to donate electrons [42]. The linoleic acid/*β*-carotene system is lipophilic and highly hydrophobic [43]. This involves determining the ability and content of lipophilic antioxidant compounds in the studied samples [44]. Both the *H. strobilaceum* and *S. fruticosa* extracts had a good amount of these antioxidant compounds (in addition to phenolic compounds, whose presence was detected through quantitative estimation).

Both *H. strobilaceum* and *S. fruticosa* extracts were able to protect cells from oxidation by hydroxyl radical. According to the Hly_50_ values, the extracts provided significant protection against hemolysis. This may be because they contain compounds that protect membrane stability by preventing the oxidation of membrane lipids, especially phospholipids, or the glycerol group [45]. There are three possible mechanisms that phenolic compounds use to protect blood cells: by directly inhibiting the hydroxyl radical, reducing Fe^3+^ ions to Fe^2+^ (stopping hydroxyl radical production) and thereby increasing the complex formation of iron-polyphenols, or by decreasing the capability of reducing H_2_O_2_ to OH^•^ [46].

Regarding the ability to reduce ferric ions, *S. fruticosa* extract showed the best antioxidant capacity with the lowest EC_50_ value (1024 ± 35 μg/mL) compared to *H. strobilaceum extract* (Table 6). The *H. strobilaceum* extract EC_50_ value obtained in this study was similar to the EC_50_ value observed by Saada et al. [38], for both apolar and polar fractions (>2000 µg/mL). The total antioxidant capacity of the *S. fruticosa* extract was 151.83 ± 2.03 mg GEA/g, indicating an equivalent antioxidant activity to gallic acid, one of the most important reducing agents found in plants [19]. The antioxidant activity of *H. strobilaceum* was lower (93.94 ± 1.92 mg GEA/mg) than that of *S. fruticosa* because it has less phenolic content (Table 4).

The six tests that were conducted proved that *H. strobilaceum* and *S. fruticosa* are characterized by the ability to reduce the lipoxidation reaction, protect against hemolysis, and scavenge free radicals, and they have a medium reducing capacity.

The preliminary antibacterial assay showed that the hydro-methanolic extract of *H. strobilaceum* had a high sensibility to Gram-negative bacteria, *E. coli*, and a weak sensibility towards *B. subtilis*, *L. innocua*, *S. typhimuruim*, *S. aureus*, and *P. aeruginosa*. These results are consistent with the quantification of anthocyanins, and consistent with the findings reported by Ma et al. [47] who showed that anthocyanin-rich extracts have a high sensibility to *E. coli*. Messina et al. [41] concluded that *H. strobilaceum* (from the Sicilian coast) has an inhibitory property against oyster pathogenic bacteria.

The *S. fruticosa* extract displayed moderate sensibility against all studied strains (9.1–15.7 mm). This observation is expected because *S. fruticosa* is known to have anti-microbial properties and is used in traditional medicine for wound healing, ophthalmology, urogenital disorders, skin diseases (scabies, herpes, dermatitis), respiratory, dental, and digestive disorders, whereas most of these diseases appear due to pathogenic microbes [48,49]. Chekroun-Bechlaghem et al. [50] confirmed that the methanol/water extracts of *S. fruticosa* leaves had potent antibacterial activity against *S. aureus* (MIC = 1.25 mg/mL). The antibacterial activity of *H. strobilaceum* and *S. fruticosa* can be attributed to their abundant content of tannins (hydrolysable and condensed tannins represented about 30 to 40% of the total phenolic compounds in the extracts). Tannins are known to be bactericidal because they irreversibly interact with proteins, thus complexing within bacterial membranes, which leads to the neutralization of their activity [51].

The ability of the extracts to prevent protein denaturation is due to their chemical content, which acts as a stabilizer for proteins. Since albumin is alkaline, the nature of the bondable compounds is acidic or moderate, such as phenolic acids [52]. RP-HPLC analysis showed that the quantitative and qualitative content of phenolic acids in *S. fruticosa* extract is more than in *H. strobilaceum* extract (Table 5). Therefore, it had the best effect, as it was equal to the pharmaceutical drug Diclofenac. The anti-inflammatory property of *S. fruticosa* was also confirmed by the results of Ksouri et al. [53]’s study, by testing on NO overproduction in LPS-stimulated RAW 264.7 macrophages.

The most important components derived from medicinal plants for the management of carcinogenesis, as an alternative source of cancer treatment, are the secondary metabolic products such as flavonoids, alkaloids, and tannins. These products do not have a cytotoxic effect on healthy cells, but have been shown to be cytotoxic to many human cancer cells [54]. The superiority of the *H. strobilaceum* extract over the *S. fruticosa* extract could be attributed to their quantitative and qualitative content of flavonoids (Table 4). Quercetin is one of the most important dietary flavonoids, and it has an antioxidant and anticancer effect [55]. Casella et al. [56] concluded that it has an antiproliferative effect because it causes cell cycle arrest and leads to programmed death in a variety of cancer cells. The results of this assay indicate that these two halophytes may be a source of anticancer compounds, particularly towards Huh-7 cells. As previously reported by Handoussa et al. [29], an extract of ethyl acetate from the aerial part of *H. strobilaceum* had potent anticancer activity against MCF-7, PC3 and A549 cells. Oueslati et al. [31] observed that the methanolic extract of *S. fruticosa* shoots had low cytotoxic activity against Detroit 551, Caco-2, HT-29, and A549 cells, and strong cytotoxic activity against colon carcinoma cell lines DLD-1. A previous investigation by Saleem et al. [49] showed that the methanol and dichloromethane extract of the same plant had low to moderate cytotoxicity against MDA-MB-231, MCF-7, and DU-145 cell lines.

## 4. Materials and Methods

### 4.1. Chemicals

Aluminum chloride (AlCl_3_), ferric chloride (FeCl₃), sodium carbonate (Na_2_CO_3_), and trichloroacetic acid were obtained from Prolabo (USA). Folin–Ciocalteu reagent, Folin–Denis reagent, and hydrogen peroxide (H_2_O_2_) were obtained from Biochem chemopharma Co. (Cosne-Courssur-Loire, France). Methanol (≥99.7% (GC), 2, 2-diphenyl-1-picrylhydrazyl (DPPH), potassium ferricyanide (K_4_Fe(CN)_6_), ascorbic acid, gallic acid, quercetin, vanillin, *p*-coumaric acid, vanillic acid, chlorogenic acid, rutin, naringenin, thiobarbituric acid (≥98%), sulfuric acid and the rest of the chemicals, reagents and organic solvents were obtained from Sigma-Aldrich (Burlington, MA, USA).

### 4.2. Plant Materials

The aerial parts of *Halocnemum strobilaceum* Pall. and *Suaeda fruticosa* L. (Figure 4) were collected during July 2022, in Chott Zebahir (34°02′05.2″ N, 6°52′47.4″ E) at −26 m altitude and Chott Amar (33°26′19.7″ N, 5°50′57.4″ E) at 42 m altitude, respectively, in the North El Oued region, Algeria. The systematic identification of the species was performed by Dr. Youcef HALIS, and a samples voucher (SSN 2 and CFN 2) was deposited in the Laboratory of Biology, University of El Oued, Algeria.

### 4.3. Determination of Physiological Indicators

#### 4.3.1. Determination of Malondialdehyde (MDA) Content

Malondialdehyde (MDA) was determined according to the method of [7]; 0.2 g of fresh plant matter was homogenized in 2 mL of 0.1% (*w*/*v*) trichloroacetic acid at 4 °C. The homogenate was centrifuged for 10 min at 1000 rpm. Then, 0.5 mL of the supernatant was taken, to which 3 mL of 0.5% (*v*/*v*) thiobarbituric acid (prepared in 20% trichloroacetic acid) was added. The mixture was incubated in a 95 °C water bath with continuous shaking for 50 min, then the samples were placed in an ice bath until the temperature was reduced to 25 °C. Samples were centrifuged for 10 min at 10,000 rpm, and the absorbance of the mixture was read at 532 nm by spectrophotometer (Jenway Model 5705 UV/visible spectrophotometer). MDA contents were calculated using the following equation:MDA level (nmol) = Δ (A_532_ nm − A_600_ nm)/1.56 × 105 (1)

#### 4.3.2. Determination of Proline Content

Proline was determined using the method of [7]; 0.5 g of fresh plant material was homogenized in 4 mL of 3% sulfosalicylic acid. The homogenates were then centrifuged for 10 min at 1000 rpm. To 1 mL of the supernatant was added 2 mL of ninhydrin acid reagent and 2 mL of glacial acetic acid in a test tube, and the mixture was incubated in a water bath at 100 °C for 60 min. After cooling, 4 mL of toluene was added to the solution mixture and vortexed. The chromium carrier containing toluene (top layer) was transferred to a new test tube. Finally, the absorbance at 520 nm was read with a spectrophotometer and toluene was used as a blank. The proline concentration was determined using the standard curve and expressed as mg/g.

#### 4.3.3. Determination of Water Content

Water content or moisture content was determined using the drying method [8], and fresh plant material was dried in an incubator at 60 °C. After every 24 h, the plant material was weighed until the weight stabilized. The percentage of moisture content was calculated according to Equation (2):(2)Water content (%)=W1−W2W1×100

W1: weight of the plant material before drying; W2: weight of the plant material after drying.

#### 4.3.4. Determination of the Content of Photosynthetic Pigments

Chlorophyll a, chlorophyll b, and carotenoids were determined by grinding 0.1 g of fresh plant material in 10 mL of acetone (80%) in a mortar, then incubating the mixture at 4 °C for 24 h. After centrifugation for 10 min at 5000× *g*, the absorbance of the supernatant was read at wavelengths 663, 645, and 470 nm using a spectrophotometer [21]. By applying Equations (3)–(5), the content of chlorophyll a, chlorophyll b, and carotenoids was calculated:Chl a (mg/mL) = 12.25 × Abs663 − 2.79 × Abs649 (3)
Chl b (mg/mL) = 21.5 × Abs649 − 5.1 × Abs663 (4)
Carotenoids (mg/mL) = (1000 × Abs470 − 1.82 × Chl a − 104.96 × Chl b)/198 (5)

### 4.4. Determination of Physicochemical Characterization

#### 4.4.1. Ash Material

The percentage of ash material was determined by applying Equation 6 as the plant material was burned at 550 °C in a in a muffle furnace (Nabertherm) [8].
(6)Ash content (%)=W1−W2W1×100

W1: weight of the plant material before burning; W2: weight of the plant material after burning.

#### 4.4.2. Determination of Mineral Elements in Plants

The weight of elements and oxides were determined using Phenom ProX desktop scanning electron microscope from Phenom-World; it is the ultimate all-in-one imaging and Energy dispersive X-ray (EDX) analysis. The results of the energy dispersive X-ray (EDX) analysis of the dry matter of the aerial part of *H. strobilaceum* and *S. fruticosa* were exploited.

#### 4.4.3. pH and Conductivity

pH and conductivity were determined by using a pH/conductivity meter (Consort™) on the solution of the plant samples (4.5 g in 45 mL of distilled water).

#### 4.4.4. Macronutrients Content (Carbohydrates, Lipids, and Proteins)

The carbohydrates, lipids, and proteins were estimated according to the method mentioned in [57]. The results are expressed as the values expressed in the equivalent of standard compounds (glucose, soybean oil, and BSA).

### 4.5. Phytochemical Study

The phytochemical study and biological tests were carried out by exploiting the hydro-methanolic extract. Extraction was carried out by soaking 20 g in 100 mL methanol: water (70:30%) for 24 h. Using rotavapor (Buchi Rotavapor R-200), the extract was concentrated.

#### 4.5.1. Quantification of Phenolic Compounds

The total estimation of phenolic compounds (TPC) was carried out according to the Folin–Ciocalteu method [58]. As for the quantitative estimation of flavonoids (TFC), this was carried out according to the colorimetric method [59], and the quantitative estimation of flavanols content (FC) by following the method described by Chekroun-Bechlaghem et al. [30]. The hydrolysable tannins content (HTC) was determined according to the Folin–Denis method [58], and the total condensed tannins (TCT) according to the vanillin-HCl method [59]. Results were expressed as µg standard compound/mg extract. The anthocyanins content (AC) was determined according to the pH differential method [19].

#### 4.5.2. RP-HPLC Analysis

The chromatographic system for the separation and analysis of phenolic acids and flavonoids were carried out with Shimadzu model prominence liquid chromatography, thermostatic column compartment, online degasser, and an UVvisible detector model SPD-20A (operating at 268 nm). The analytical column used was a Shim-pack VP-ODS C18 (4.6 mm × 250 mm, 5 μm) (Shimadzu Co., Kyoto, Japan). A binary gradient linear system consisting of acetonitrile (A) and 0.2% acetic acid in water (B) was used. The gradient method was generated by starting with 90% B, then decreasing to 86% B in 6 min, to 83% B in 16 min, to 81% B in 23 min, to 77% B in 28 min, holding at 77% B at 28–35 min, then decreasing to 60% B in 38 min, to 90% B in 50 min at a flow rate of 1 mL/min. The quantification of separated peaks was performed by calibration with standard gallic acid, chlorogenic acid, vanillic acid, caffeic acid, *p*-coumaric acid, vanillin, rutin, naringin, and quercetin. The phenolic composition was quantified by plotting a standard curve with the respective standards. The chromatographic method used in this study had been previously investigated and validated by one of the current authors in a previous study [37].

### 4.6. Biological Activity

#### 4.6.1. Antioxidant Activity

DPPH^•^ and HO^•^ scavenging activity: the DPPH^•^ test was applied as mentioned by Muthukrishnan et al. [59]. The hydroxyl radical scavenging test was mentioned by Saeed et al. [60].

*β*-carotene-linoleic acid assay: antioxidant activity was evaluated using the *β*-carotene bleaching method as described by Chekroun-Bechlaghem et al. [30].

Determination of anti-hemolysis activity: the efficacy of the extracts in protecting red blood cells (RBCs) from demolition by exogenous oxidants was evaluated following the protocol described by Chouikh et al. [57].

Reducing power was determined by the ferric reducing antioxidant power assay and the phosphomolybdate test according to Muthukrishnan et al. [59].

#### 4.6.2. Evaluation of Antibacterial Activity

The sensibility of six strains of bacteria (*Bacillus subtilis*, *Listeria innocua*, *Escherichia coli*, *Pseudomonas aeruginosa*, *Salmonella typhimurium*, and *Staphylococcus aureus*) was tested using the Disk Diffusion method [59]. The bacterial suspension was grown on Mueller-Hinton agar, and then discs (Whatman paper, 6 mm in diameter, 0.3 mm in height) containing a volume of 10 μL of extracts (4 mg) were spread. After incubation for 24 h at 37 °C, the results were noted by measuring the diameter of the inhibition zone, which indicated that the bacteria were sensitive, moderately sensitive, or resistant to the antibiotic being tested. All steps were performed under sterile conditions. DMSO was used as a negative control as it is the solvent used to dissolve the extracts. Gentamicin (1 mg) was used as a positive control in the assay.

#### 4.6.3. In Vitro Anti-Inflammatory Activity

The assay method was carried out according to [57]. Diclofenac sodium was used as a reference drug. Briefly, 1 mL of serum albumin (5%), 1 mL of different concentrations of the studied samples, and 20 μL of hydrochloric acid (1 N) were mixed. The mixture was then incubated in the incubator at 37 °C for 20 min and then placed in a water bath at 57 °C for three minutes. After cooling, 2.5 mL of phosphate-buffered solution (0.1 M, pH = 6.4) was added. Absorbance was measured at 660 nm. The percentage protein protection against denaturation was calculated using the following formula:Percentage protection against denaturation = [(1 − Abs_Sample_)/Abs_Control_] × 100 (7)

#### 4.6.4. Cytotoxicity Assay

Cancer cells were cultured at 37 °C and 5% CO_2_ atmosphere (*v*/*v*) in Dulbecco’s Modified Eagle Medium, plus 10% of heat-inactivated serum, and the antibiotic (100 mg/mL streptomycin and 100 units/mL penicillin).

##### Huh-7 Cell Viability Was Assessed by WST-1 Assay

According to method [61], 96-well plates were incubated for 24 h and contained aliquots of 50 µg/mL cell suspension (3 × 10^3^ cells). Then, another aliquot of 50 µL media containing the studied samples was added at gradient concentrations (0.01–100 µg/mL) and left for 48 h, then the cells were treated with 10 µL of the cell proliferation reagent (WTS-1) and incubated for one hour. Absorbance was measured at 450 nm.

##### HepG2 Cell Viability Was Assessed by SRB Assay

Aliquots of 100 μL of cell suspension (5 × 10^3^ cells) were incubated in complete media for 24 h. Then, the cells were treated with another aliquot of 100 μL of media containing the samples at different concentrations. After 72 h, media were replaced with 150 μL of trichloroacetic acid (10%) and incubated at 4 °C for one h. Then, the trichloroacetic acid solution was removed, and cells were washed five times with distilled water. Aliquots of 70 μL of sulforhodamine B (SRB) (0.4%) solution were then added and incubated in a dark place at room temperature for 10 min. The plates were then washed three times with acetic acid (1%) and left to air dry. Then, 150 μL of TRIS (10 mM) was added. The absorbance was measured at 540 nm [62]. Applying the following relationship, the viability percentage is calculated:(8)$$Percentage\;of\;viability\;of\;cell=\frac{{Abs}_{Sample}}{{Abs}_{Control}}\times 100$$

Epiphany was used as a positive control.

### 4.7. Statistical Analysis

The data were expressed as the mean ± standard deviation of three replicates. The statistical study was carried out using SPSS Statistic for Windows, SPSS Statistics for Windows, version 15.0 (IBM, Chicago, IL, USA). The differences between the two variables were determined using Student’s *t*-test for quantitative estimates. For biological activity data, a one-way ANOVA test (LSD) was used.

## 5. Conclusions

The data of this study prove that the highly saline environmental conditions in which *Halocnemum strobilaceum* and *Suaeda fruticosa* are found made them physiologically tolerant plants capable of accumulating an important mineral content and achieving an osmotic balance with the soil solution. It is possible that this could be an important source of minerals, especially for calcium. Through its strategy to escape from these conditions with minimal damage, it made it a source of primary metabolites (proteins, carbohydrates, and lipids). Thus, it became capable of being introduced into all food industries or as part of supplementary feeding for livestock. These conditions also achieved oxidative stress events for two plants, which resulted in their good content, both quantitatively and qualitatively, of phenolic compounds. Through investigations on the therapeutic use of *H. strobilaceum* and *S. fruticosa*, it has been shown that it is acceptable as a promising source of anti-inflammatory and antioxidant, antimicrobial, and cytotoxic properties. However, it would be preferable to complete the in vivo study, and it is necessary to study the safety and toxicological aspects of these two plants, especially because they are rich in oxides.

## Figures and Tables

**Figure 1 molecules-28-03580-f001:**
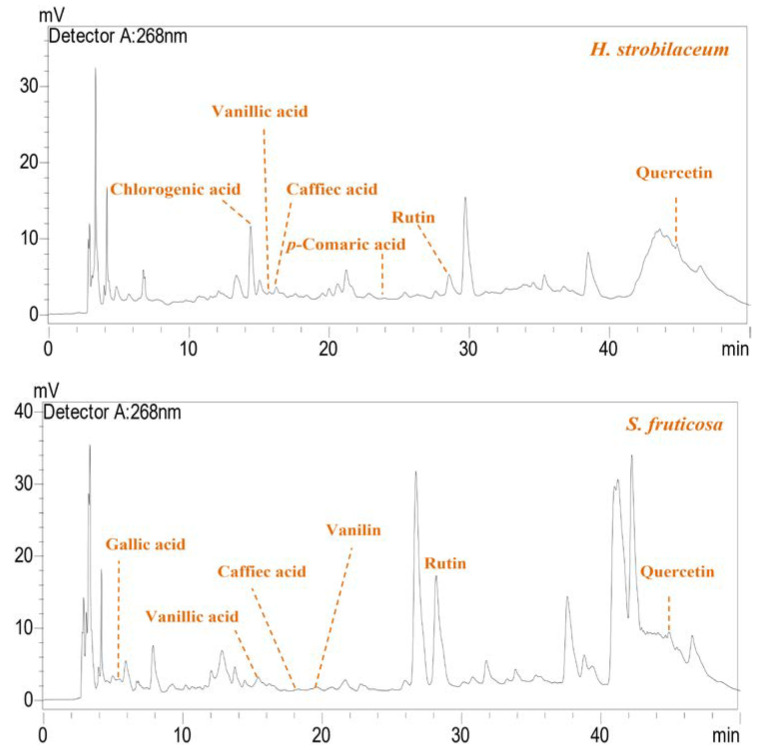
RP-HPLC chromatogram of the identified phenolic compounds in the hydro-methanolic extracts (*H. strobilaceum* and *S. fruticosa*).

**Figure 2 molecules-28-03580-f002:**
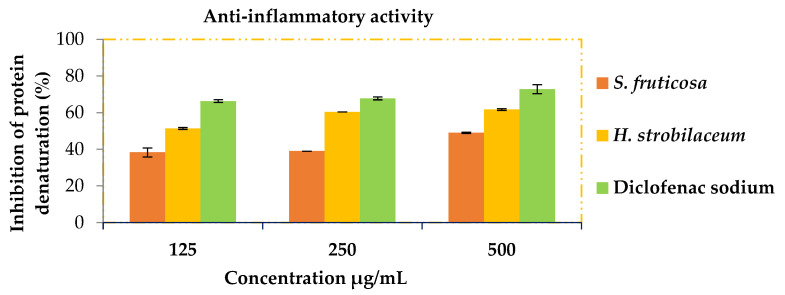
The anti-inflammatory activity of *H. strobilaceum*, *S. fruticosa* and the positive control (Diclofenac sodium) (125, 250, and 500 μg/mL).

**Figure 3 molecules-28-03580-f003:**
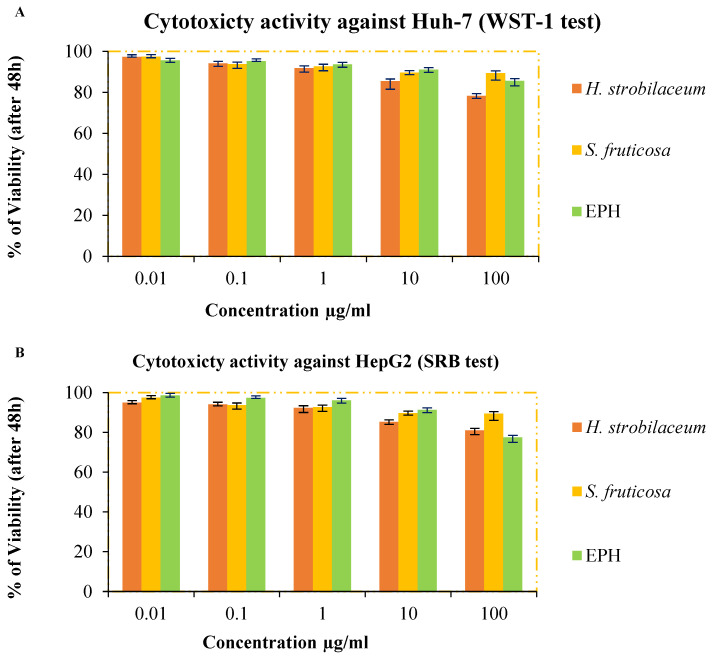
In vitro cytotoxicity assay for the hydro-methanolic extracts of *H. strobilaceum*, *S. fruticosa* and the positive control (EPH) against Huh-7 (**A**) and HepG2 (**B**) cells (0.01, 0,1, 1, 10, and 100 μg/mL).

**Figure 4 molecules-28-03580-f004:**
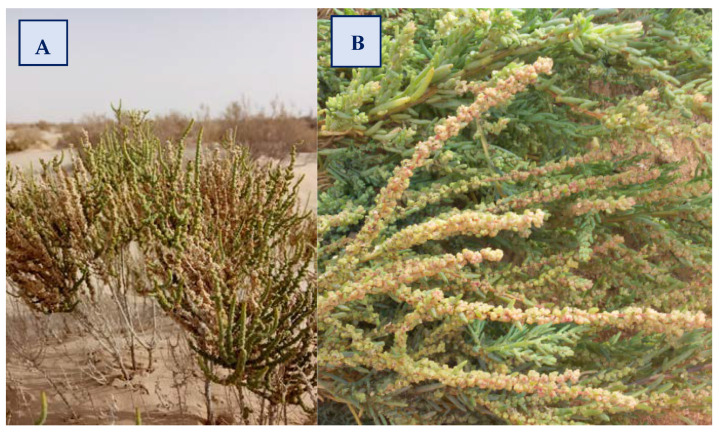
Plant material. Morphological characters of the aerial parts of two halophytes, (**A**) *Halocnemum strobilaceum* Pall and (**B**) *Suaeda fruticosa* L.

**Table 1 molecules-28-03580-t001:** Physiological statements of the aerial part of *H. strobilaceum* and *S. fruticosa*.

	Moisture Content (%)	Proline (mg/g)	MDA (mM)	Ch a (mg/g)	Ch b (mg/g)	Carotene (mg/g)
*H. strobilaceum*	46.94 ± 5.04	0.366 ± 0.06	0.24 ± 0.06	2.79 ± 0.01	0.6 ± 0.02	0.88 ± 0.02
*S. fruticosa*	82.7 ± 3.56	0.434 ± 0.09	0.12 ± 0.02	3.22 ± 0.01	0.91 ± 0.03	0.99 ± 0.00

**Table 2 molecules-28-03580-t002:** SEM-EDX analyses for the dry matter of the aerial part of *H. strobilaceum* and *S. fruticosa*.

	Weight (%)														
	Elements										Oxides				
	O	C	Ca	Na	Si	Al	Cl	Mg	K	S	CaO	Na_2_O	SiO_2_	Al_2_O_3_	MgO
*H. strobilaceum*	43.05	32.22	3.23	7.51	1.11	0.41	6.15	0.17	0.42	0.86	6.61	21.3	3.67	1.31	0.48
*S. fruticosa*	25.04	49.55	1.59	1.92	0.49	0.3	6.32	0.75	5.16	0.25	4.46	1.57	2.1	1.14	3.53

**Table 3 molecules-28-03580-t003:** The ash, pH/electrical conductivity, and macronutrients content (carbohydrates, lipids, and proteins) of *H. strobilaceum* and *S. fruticosa*.

	Ash (%)	pH	EC (dmS/m)	CCA (mg G/g)	LC (mg SO/g)	PC (mg BSA/g)
*H. strobilaceum*	29.89 ± 1.8	7.5 ± 0.005	19.4 ± 0.00	8.13 ± 0.17	2.68 ± 0.02	15.2 ± 0.04
*S. fruticosa*	19.76 ± 1.21	6.15 ± 0.09	12.38 ± 0.07	9.07 ± 0.22	2.25 ± 0.15	15.6 ± 0.09

**Table 4 molecules-28-03580-t004:** Content of phenolic compounds in the hydro-methanolic extracts from the aerial part of *H. strobilaceum* and *S. fruticosa*.

	TPC (µg GAE/mg)	TFC (µg QE/mg)	FC (µg QE/mg)	AC (µg C-3-GE/mg)	HTC (µg GAE/mg)	TCT (µg CE/mg)
*H. strobilaceum*	24.97 ± 0.09	12.17 ± 0.16	5.43 ± 0.06	1.87 ± 1.88	6.23 ± 0.24	3.99 ± 0.09
*S. fruticosa*	47.38 ± 0.16	14.57 ± 0.12	6.70 ± 0.16	1.17 ± 0.47	8.81 ± 0.32	4.68 ± 0.25

**Table 5 molecules-28-03580-t005:** Content of individual phenolic acids and flavonoids in the hydro-methanolic *H. strobilaceum* and *S. fruticosa* extracts.

Compounds		Retention Time (min)	*H. strobilaceum*	*S. fruticosa*
Phenolic acid (µg/100 mg ED)	Gallic acid	5.29	-	10.01
	Chlorogenic acid	13.392	85.77	-
	Vanillic acid	15.531	1.33	20.30
	Caffeic acid	16.277	2.71	5.66
	Vanillin	21.46	-	17.36
	*p*-Coumaric acid	23.817	0.96	1.63
Flavonoide (µg/100 mg ED)	Rutin	28.37	84.42	367.56
	Naringin	34.788	-	-
	Quercetin	45.047	207.16	93.69

**Table 6 molecules-28-03580-t006:** IC_50_, EC_50_, or Hly_50_ values (µg/mL) for methods of evaluating antioxidant activity.

			*H. strobilaceum*	*S. fruticosa*	Ascorbic Acid	α-Tocopherol
Radical scavenging activity	DPPH^•^	IC_50_	81.70 ± 0.64	118.8 ± 1.46	1.44 ± 0.02	/
	HO^•^	IC_50_	>1000	>1000	86.0 ± 0.70	/
*β*-carotene bleaching method		EC_50_	58.8 ± 0.94	82.8 ± 2.23	532.4 ± 2.50	2.10 ± 0.08
Anti-hemolysis activity		Hly_50_	193.3 ± 1.70	225.7 ± 27.80	154.4 ± 1.70	/
Reducing power		EC_50_	>2000	1024 ± 35	67.28 ± 2.00	/
Total antioxidant capacity (mg GAE/g)			93.94 ± 1.92	151.83 ± 2.03	/	/

**Table 7 molecules-28-03580-t007:** Screening antibacterial activity of the *H. strobilaceum* and *S. fruticosa* hydro-methanolic extracts.

Treatments (mg)	Inhibition Zone Diameter (mm)					
	Gram Positive			Gram Negative		
	*B. subtilis* ATCC-6633	*L. innocua* CLIP-74915	*S. aureus* ATCC-6538	*E. coli* ATCC-25922	*P. aeruginosa* ATCC-9027	*S. typhimurium* ATCC-14028
*H. strobilaceum* (4 mg)	10.7 ± 1.2 *	11.3 ± 4.7 *	11.3 ± 4.7 *	15.3 ± 3.2 **	8 ± 1.7	11.7 ± 1.2 *
*S. fruticosa* (4 mg)	9.7 ± 0.6	14.3 ± 1.5 *	14.3 ± 1.5 *	9.7 ± 0.6	12.3 ± 4 *	11.7 ± 0.6 *
Gentamicin (1 mg)	10.67 ± 2.1 *	10.67 ± 2 *	11.33 ± 1.1 *	/	30.33 ± 0.6 ***	14.33 ± 0.58 *

The antibacterial activity was classified as follows: zone diameter: ≥21 mm, strong sensibility ***; 15–20 mm, moderate sensibility **; 10–15 mm, weak sensibility *; and ≤10 mm, little or no sensibility.

## Data Availability

Not applicable.

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
