# Peer review of "Biochemical Profile and In Vitro Therapeutic Properties of Two Euhalophytes, Halocnemum strobilaceum Pall. and Suaeda fruticosa (L.) Forske., Grown in the Sabkha Ecosystem in the Algerian Sahara"

_molecules, 2023, doi:10.3390/molecules28083580_

Round 1

Reviewer 1 Report

General comment:

The biochemical profile of H. strobilaceum and S. fruticosa were investigated in this manuscript, and the biological activities including anti-inflammatory, andtibacterial, antioxidant and cytotoxic properties were also studied. This manuscript is of interest and offers a worthy contribution to the field. However, some revisions are required. The following comments should help further improve the quality of the work:

Specific comment:
1. Result: There's no statistical analysis in some data. It is always good to have some statistical analysis in the results as it will strengthen the discussion.

2. Discussion: Kindly improve on the discussion. What is the significance of the results of the work?

3. Material and Method:
3.1- The source and purity of materials used need to be specified.
3.2- The manufacturer, model of the equipment used are also encouraged to be stated.

4. References
4.1- Kindly revise reference format according to the author guideline.
4.2- It is suggested to cite references within 5 years of research to maintain the reliability of results obtained.
4.3- There are references found to be outdated.

Author Response

April 06th 2023

Journal: Molecule

Manuscript Number: Molecule-2322542

Title:  Biochemical profile and in vitro therapeutic properties of two euhalophytes; Halocnemum strobilaceum Pall. and Suaeda fruticosa (L.) Forske. grown in the Sabkha ecosystem in the Algerian Sahara

Dear editor,

We gratefully appreciate the rigorous review of the referees who reviewed our manuscript, which added much strength and validity to our research. Some points really helped to improve the manuscript and we really appreciate that. Again, we thank all the reviewers for their valuable inputs, which have given the manuscript a chance to reach a satisfactory level for publication.

(Added words correlated with the answers are marked in yellow in the revised manuscript)

Our responses are as follows:

Reviewer #1

Comment 1

- General comment:

The biochemical profile of H. strobilaceum and S. fruticosa were investigated in this manuscript, and the biological activities including anti-inflammatory, andtibacterial, antioxidant and cytotoxic properties were also studied. This manuscript is of interest and offers a worthy contribution to the field. However, some revisions are required. The following comments should help further improve the quality of the work:

 Response 1

We would like to thank the reviewer for this comment and thorough reading of this manuscript and for the thoughtful comments and constructive suggestions. Under the reviewer’s suggestion, we have corrected and improved our manuscript.

Comment 2

- Specific comment:

  1. Result: There's no statistical analysis in some data. It is always good to have some statistical analysis in the results as it will strengthen the discussion.

Response 2

Thank you, we are really sorry for this error. Under the reviewer’s suggestions, we have checked and corrected the statistical analysis of all our data.

Comment 3

  1. Discussion: Kindly improve on the discussion. What is the significance of the results of the work?

Response 3

We thank the reviewer for this comment, We are really sorry for this error. Under the reviewer’s suggestions, it was corrected. (Please see line 248 to 252 page9)

Comment 4

  1. Material and Method:

3.1- The source and purity of materials used need to be specified.

Response 4

Under the reviewer’s suggestions, we have checked and corrected (Please see line  432 to 439 page 13)

Comment 5

  1. Material and Method:

3.2- The manufacturer, model of the equipment used are also encouraged to be stated.

Response 5

We thank the reviewer for this remark, we are really sorry for this error, Under the reviewer’s suggestions, we have checked and corrected.

Comment 6

  1. References

4.1- Kindly revise reference format according to the author guideline.

4.2- It is suggested to cite references within 5 years of research to maintain the reliability of results obtained.

4.3- There are references found to be outdated.

Response 6

We would like to thank the reviewer for their remarks and suggestions. Under the reviewer’s suggestion, we have corrected and improved the references in this revised version manuscript and we have added some important references related to this study within the last 5 years.

We hope the Reviewer and the Editors will be satisfied with our responses to the ‘comments’ and the revisions for the original manuscript.

 Thanks and Best Regards!

 Yours Sincerely,

Reviewer 2 Report

Regarding HPLC chromatography, how accurate is it?

Regarding the cytotoxicity test, was there a statistically significant difference?

Author Response

April 07th 2023

Journal: Molecule

Manuscript Number: Molecule-2322542

Title:  Biochemical profile and in vitro therapeutic properties of two euhalophytes; Halocnemum strobilaceum Pall. and Suaeda fruticosa (L.) Forske. grown in the Sabkha ecosystem in the Algerian Sahara

Dear editor,

We gratefully appreciate the rigorous review of the referees who reviewed our manuscript, which added much strength and validity to our research. Some points really helped to improve the manuscript and we really appreciate that. Again, we thank all the reviewers for their valuable inputs, which have given the manuscript a chance to reach a satisfactory level for publication.

(Added words correlated with the answers are marked in yellow in the revised manuscript)

Our responses are as follows:

Reviewer#2

Comment 1

Regarding HPLC chromatography, how accurate is it?

 Response 1

At first, we would like to thank the reviewer for this comment and thorough reading of this manuscript and for the thoughtful comments and constructive suggestions.

Secondly, concerning the reviewer's question regarding the accuracy of the HPLC chromatography used in this study, the accuracy of HPLC analysis can vary depending on various factors, including the quality and purity of the analytical standards used, the precision and accuracy of the instrument, and the expertise of the operator. However, in the case mentioned in the passage, the method's accuracy was previously validated by one of the authors in a previous study, and the reference was provided in the corrected version. If further details about the method are required, a supplementary material containing the method's details can be added. Overall, HPLC can be highly accurate and reliable when performed correctly and validated appropriately.

Comment 2

Regarding the cytotoxicity test, was there a statistically significant difference?

Response 2

We thank the reviewer for this comment and for highlighting these points regarding the cytotoxicity test, and for the statistical significance, our answer is yes, we have add more discussion about that. (Please see line 169 to 171 page 6)

We hope the Reviewer and the Editors will be satisfied with our responses to the ‘comments’ and the revisions for the original manuscript.

 Thanks and Best Regards!

 Yours Sincerely,

Reviewer 3 Report

Dear Authors,

Halocnemum strobilaceum Pall. and Suaeda fruticosa L. are plants that have been previously studied in terms of content and activity, it is important to clearly state the novelty of this study and to compare the data obtained in this study with previous studies. In order to emphasize the regional difference of these plants, it is necessary to give a comparison with the data obtained from other regions for the same plants.

 It would be more appropriate to give Figures 1,2 and 3 as tables.

It is recommended that the graphics in Figure 4 be larger. The graph given in Figure 5 in the Results section should be reduced to Figure 6 size. For Figure 5 given in the Materials and Methods section, a different number and figure subtitle should be specified more descriptively.

Thanks

Author Response

April 06th 2023

Journal: Molecule

Manuscript Number: Molecule-2322542

Title:  Biochemical profile and in vitro therapeutic properties of two euhalophytes; Halocnemum strobilaceum Pall. and Suaeda fruticosa (L.) Forske. grown in the Sabkha ecosystem in the Algerian Sahara

Dear editor,

We gratefully appreciate the rigorous review of the referees who reviewed our manuscript, which added much strength and validity to our research. Some points really helped to improve the manuscript and we really appreciate that. Again, we thank all the reviewers for their valuable inputs, which have given the manuscript a chance to reach a satisfactory level for publication.

(Added words correlated with the answers are marked in yellow in the revised manuscript)

Our responses are as follows:

Reviewer #3

Comment 1

Dear Authors,

Halocnemum strobilaceum Pall. and Suaeda fruticosa L. are plants that have been previously studied in terms of content and activity, it is important to clearly state the novelty of this study and to compare the data obtained in this study with previous studies.

Response 1

We would like to thank the reviewer for this comment and through the reading of this manuscript and for the thoughtful comments and constructive suggestions.

So, let us dear respect the reviewer to clarify that, these plants Halocnemum strobilaceum Pall. and Suaeda fruticosa L., have many and varied uses by Algerian peoples, and to our knowledge; no information and study exist on these plants collected from Algerian Sahara, about the Biochemical profile and therapeutic properties.

So, the data obtained in this investigation on these plants will be helpful to the Algerian pharmaceutical industry and medical research.

Author hand,

These plants were collected from From the desert in the southeast of Algeria, ( original wild-growing in Algerian sahara) and the originality of the work is, the authenticity of this plant study at the place of harvest, as well as the climate (semi-arid region of Algeria).

Yes, there are some others studies about the mineral concentration and anti-inflammatory priority in this plant in other countries, but the proportions of these compounds differ from region to another. In addition, these previous studies attributed the reasons for this difference to the difference in climate (rainy-moderate-dry), soils, and altitude above sea level, as well as the harvest period (before-after) flowering , and also age of the plant (first-second- third harvest).

The data obtained in this work can be used as a scientific source of information for pharmaceutical and medical research about these plants Halocnemum strobilaceum Pall. and Suaeda fruticosa (L.) Forske., grown in the Sabkha ecosystem in the Algerian Sahara

Comment 2

- In order to emphasize the regional difference of these plants, it is necessary to give a comparison with the data obtained from other regions for the same plants.

Response 2

Thank the reviewer for this comment, under the reviewer's suggestion, we added in the revised manuscript some paragraphs and one table about the comparison of our data with previously reported findings, we compared our data, with the literature. (Please see line 292 and 308 page 19)

Comment 3

It would be more appropriate to give Figures 1,2 and 3 as tables.

Response 3

We thank the reviewer for this comment, We are really sorry for this error. Under the reviewer’s suggestions, it was corrected. (Please see Table 3 which replaced Figure 1 and Table 4 which replaced Figure 2, (line 114, and Line 121 page 4))

Comment 4

It is recommended that the graphics in Figure 4 be larger.

Response 4

Thank the reviewer for this remark, and we are very sorry for our unclear report about these. Under the reviewer's suggestion, we have improved and larger Figure 4, which now is very clear. (Please see line 133 page 4 and 5)

Comment 5

The graph given in Figure 5 in the Results section should be reduced to Figure 6 size.  

Response 5

Acknowledgment, under the reviewer's suggestion, we have checked and corrected Figure 5.

Comment 6

For Figure 5 given in the Materials and Methods section, a different number and figure subtitle should be specified more descriptively.

Response 6

We thank the reviewer for this comment, We are really sorry for this error, and it was corrected.

We hope the Reviewer and the Editors will be satisfied with our responses to the ‘comments’ and the revisions for the original manuscript.

 Thanks and Best Regards!

 Yours Sincerely,

Round 2

Reviewer 3 Report

Dear Authors,

Thanks for the revisions.

Wish you all success in your studies.